# Evaluation of Nitrogen and Carbon Stable Isotopes in Filter Feeding Bivalves and Surficial Sediment for Assessing Aquatic Condition in Lakes and Estuaries

**James L. Lake [1],\*, Jonathan R. Serbst [1], Anne Kuhn [1] , Michael Charpentier [2] and Nathan J. Smucker [3]**

[1] US Environmental Protection Agency, Office of Research and Development, 27 Tarzwell Drive, Narragansett, RI 02882, USA
[2] General Dynamics Information Technology, 27 Tarzwell Drive, Narragansett, RI 02882, USA
[3] US Environmental Protection Agency, Office of Research and Development, 26 Martin Luther King Drive W., Cincinnati, OH 45268, USA
\* Correspondence: lake.jim@epa.gov

**Abstract:** Excessive inputs of nitrogen from anthropogenic activities in watersheds can cause detrimental effects to aquatic ecosystems, but these effects can be difficult to determine based solely on nitrogen concentrations because of their temporal variability and the need to link human activities to ecological responses. Here, we (1) tested the use of stable isotopes of nitrogen ($\delta^{15}N$) and carbon ($\delta^{13}C$) in benthic organic matter (BOM) as proxies for isotope ratios of filter feeding bivalves in lakes and estuaries, which can be used as indicators but are harder to sample and often spatially sparse, and (2) evaluated if stable isotope ratios in benthic organic matter could be used to assess impacts from anthropogenic land development of watersheds. The $\delta^{15}N$ in BOM isolated from surficial sediment ($\delta^{15}N_{BOM}$) was significantly correlated with $\delta^{15}N$ in filter feeding unionid mussels (*Elliptio complanata*, $\delta^{15}N_{UN}$) from lakes and with hard-shell clams (*Mercenaria mercenaria*, $\delta^{15}N_{MM}$) from estuaries. In lakes, $\delta^{13}C_{BOM}$ was significantly correlated with $\delta^{13}C_{UN}$, but $\delta^{13}C_{BOM}$ was not significantly correlated with $\delta^{13}C_{MM}$ in estuaries. Values of $\delta^{15}N_{BOM}$ and $\delta^{15}N_{UN}$ were significantly and positively correlated with increasing amounts of impervious surface, urban land cover, and human populations in watersheds surrounding lakes. In estuaries, $\delta^{15}N_{BOM}$ was only significantly and positively correlated with greater percent impervious surface in the watersheds. Correlations of $\delta^{13}C_{BOM}$ in lakes and estuaries, $\delta^{13}C_{UN}$, and $\delta^{13}C_{MM}$ with land use and human population were mostly non-significant or weak. Overall, these results show that $\delta^{15}N_{BOM}$ can serve as a proxy for $\delta^{15}N$ of filter feeding bivalves in lakes and estuaries and is useful for assessing anthropogenic impacts on aquatic systems and resources. Our study area was limited in size, but our results support further studies to test the application of this sediment stable isotope-based technique for assessing and ranking aquatic resources across broad geographical areas.

**Keywords:** nitrogen; nutrients; impervious surface; watershed; urban; sediments; bivalves; nitrogen and carbon stable isotopes





## 1. Introduction

Large monitoring programs have found that nutrient over-enrichment, often resulting from human wastewater and agricultural and urban runoff, is a leading cause of degradation of environmental conditions in aquatic systems across the United States [1–3]. The extent and magnitude of nitrogen loading to aquatic ecosystems are of particular concern because they are linked to species loss, significant degradation of water quality, eutrophication, and the development of harmful algal blooms [4–6]. These negative effects of nutrient pollution extend from watersheds to coastal and estuarine ecosystems that are commonly nitrogen limited [7–9]. The condition of lake and estuarine ecosystems can be difficult to determine based on nitrogen concentrations alone because of their temporal variability and

the need to link concentrations to ecological effects [10–13]. Identifying how organisms are associated with watershed conditions and how they can be used as indicators provides context for understanding and managing nutrient effects, especially because they integrate nutrient conditions over time [14–16].

To provide a time integrated metric of nitrogen pollution in aquatic sites, studies have identified relationships between stable isotopes of nitrogen in aquatic organisms and variables indicative of anthropogenic land development in the surrounding watersheds [17–19]. Relationships like these provide information on ecological effects of nitrogen, can be used to develop indicators for use in monitoring programs, and can help inform nutrient management efforts. Filter feeding bivalves, such as mussels, are widely used for inferring trophic baselines in streams, rivers, and lakes because of their long lifetimes and because their $\delta^{15}N$ ratios change depending on the mix of nitrogen inputs from human wastewater, atmospheric deposition, and agricultural fertilizers entering aquatic sites [20–24]. Recently, the $\delta^{15}N$ ratios of chironomids were found to be useful for understanding nitrogen inputs, processing, and transformations in streams and lakes as well, and their $\delta^{15}N$ ratios were associated with landscape, water body, and biological factors [25]. In saltwater, $\delta^{15}N$ ratios in both soft tissues and shells of hard-shell clams (*Mercenaria mercenaria*, MM) have been found to directly relate to the percentage of wastewater input to small estuaries [26,27]. Overall, these studies support the use of $\delta^{15}N$ in these organisms as a time integrative measure of human impact; however, the use of chironomids, which are almost exclusively freshwater taxa, and the use of filter feeding bivalves as indicators can be limited by the availability of organisms and constrained by the need to compare among species with potentially differing feeding strategies and physiologies [25,28]. These limitations highlight the need for a readily available ecological indicator for assessing aquatic condition in lakes and estuaries.

The $\delta^{15}N$ in benthic organic material (BOM) ($\delta^{15}N_{BOM}$) isolated from surficial sediments is one sample type potentially useful as a baseline; especially because of its nearly universal availability in lakes and estuaries. $\delta^{15}N_{BOM}$ can be significantly correlated with $\delta^{15}N$ of organisms at different trophic positions despite BOM being composed of a mixture of detritus, benthic microbes, and deposited living and dead plankton from the water column. For example, significant relationships were found between $\delta^{15}N_{BOM}$ and $\delta^{15}N$ of fish and mussels collected from lakes representing a wide range anthropogenic impact [22,28]. Additionally, significant relationships have been reported between $\delta^{15}N$ in sedimented organic material from surface sediments of tidal flats and $\delta^{15}N$ in manila clams, (*Ruditapes philippinarum*) among sites spanning a wide range of eutrophication levels [29]. In another study, $\delta^{15}N$ ratios in two species of bivalves from estuaries on Cape Cod, USA had significant relationships with $\delta^{15}N$ in organic material from sediments [30]. The use of $\delta^{15}N$ in sediment as an indicator is further supported by studies that found a correspondence of $\delta^{15}N$ ratios between sediment and seston [31] and findings that surface sediment organic matter recorded the weighted mean isotopic ratio produced in surface waters [32]. Carbon stable isotopes ($\delta^{13}C$) provide additional information that complements $\delta^{15}N$ results by establishing a baseline value for food chains and for calculation of organismal trophic positions in aquatic systems [21,28,33,34].

With multiple studies showing the potential usefulness of relationships between $\delta^{15}N_{BOM}$ and $\delta^{15}N$ of other organisms, we surveyed 51 lakes and 28 sites in nine estuaries to document and compare these relationships among sites in these two distinct ecosystems and to examine their possible use as indicators of the effects of anthropogenic disturbance in their watersheds. Our goals were: (1) to determine if $\delta^{15}N_{BOM}$ and $\delta^{13}C_{BOM}$ sampled from estuaries and lakes represented corresponding isotopic ratios in filter feeding bivalves (primary consumers), and (2) to evaluate the use of $\delta^{15}N$ and $\delta^{13}C$ in BOM and filter feeding bivalves as indicators of human impact on aquatic systems resulting from anthropogenic land development in watersheds. Documenting these relationships is important to developing the use of $\delta^{15}N_{BOM}$ as an indicator of nitrogen effects, which could be particularly useful because of the widespread availability of BOM in lakes and estuaries. Results from this study can be useful to monitoring programs interested in identifying effects of nutri-

ent pollution and for informing decisions and management strategies seeking to protect aquatic resources and to improve those experiencing negative effects of eutrophication from human sources. Results from the present study also may provide support for the use of stable isotopes in BOM for national assessments and rankings of environmental conditions in estuaries [3] and, if appropriate, as an addition to other approaches (e.g., use of chironomids) in future assessments in lakes, streams, and rivers [1,2].

## 2. Methods

Lakes and estuaries were selected to represent a wide range of anthropogenic land development of surrounding watersheds. Single samples of BOM from surficial sediments were collected from littoral zones of 51 lakes in Rhode Island, USA. These lakes had small surface areas, ranging from 3.4 to 430 ha, but were within watersheds that ranged from 23 ha to 2250 ha for lakes that were part of larger river systems (Supplementary Table S1). Unionid (UN) mussels (*Elliptio complanata*) were collected from the littoral zones by hand or with a clam rake within ~5 m of where the sediment was sampled for BOM, but they were observed and sampled in only 26 of the 51 lakes. Triplicate samples of BOM from surficial sediments were collected from a total of 28 sampling stations along the shorelines of four Rhode Island and five Massachusetts estuaries, which had watersheds ranging from 250 ha in the smallest salt pond to 275,000 ha for Narragansett Bay. Hard shell clams (*Mercenaria mercenaria*; MM) were collected with a clam rake within a ~10 $m^2$ area around the location of the BOM sample.

Bivalve tissue samples were from a section of the foot of UN and MM. These individual samples were oven dried at 60° C for >2 days, ground and weighed for isotope ratio mass spectrometry (IRMS) analysis. Each BOM sample and bivalve sample was analyzed individually, and means, or individual $\delta^{15}N_{UN}$ and $\delta^{13}C$ values where only one bivalve was found, were used in comparisons. We report the number and sizes of bivalve samples found at sample sites in Supplementary Tables S2 and S3.

Sediment samples in both lakes and estuaries were collected with a hand-held piston coring sampler, which obtained a 6.5 cm diameter core. In both lakes and estuaries, samples were taken by wading where water depths were <1 m. The corer was driven into the sediment, which varied from sandy to highly silty. The corer was withdrawn and, while maintained vertically, a core cap was pushed onto the bottom of the core tube, which was then detached from the sampler. Samples of benthic organic material (BOM) were obtained from the surface layer (~the top 1.5 to 2.0 cm) of the intact sediment core. The surface layer was re-suspended by stirring, and the suspended material was removed with a 30 mL pipette. Samples were placed in clean plastic bottles, capped, placed on ice, and transported to the laboratory. Samples were refrigerated (4 °C) until they were processed within 96 h of sampling. Samples were stirred, poured through a 0.5 mm Nytex® screen (to remove large plant material, leaves, twigs, gravel, etc.) and collected in 55 mL centrifuge tubes. Samples were shaken and centrifuged at $1000 \times g$ for five minutes. The supernatant was poured off and a spatula was used to take samples from the top ~1.5 cm of the material in the bottom of the tube. Samples were dried in an oven at 40 °C for at least three days, ground with mortar and pestle to a fine powder, weighed to 0.01 mg on a microbalance, and analyzed using an Isotope Ratio Mass Spectrometer (IRMS).

The nitrogen and carbon isotopic composition of the BOM samples and tissue samples were determined by IRMS using a Vario Micro Elemental Analyzer interfaced to an Elementar Isoprime 100 Mass Spectrometer (Elementar Americas, Mt. Laurel, NJ, USA). The nitrogen isotopic composition of the samples is expressed as a part per thousand deviation ($\delta^{15}N$ ‰) from that of the reference material, which was $N_2$ in air. Carbon isotopic composition is expressed as a part per thousand deviation ($\delta^{13}C$ ‰) from the reference standard Pee Dee Belemnite. All samples were analyzed in duplicate with a typical difference of about 0.1‰ and reported as mean values. An examination of results from wheat flour taken through all steps of the sample preparation procedure with original unprocessed flour

showed no appreciable change in $\delta^{15}$N or $\delta^{13}$C ratios demonstrating that no contamination of samples resulted from sample preparation.

We were concerned that collection of BOM samples might also include shell fragments, and if so, that the elevated $\delta^{13}$C of carbonate ($CaCO_3$) in mollusk shells [35] and in sediment might be problematic for using $\delta^{13}C_{BOM}$ for comparisons with $\delta^{13}$C in bivalves, and for relating to land cover variables. In lakes, we conducted a pilot study to determine whether removal of inorganic carbon in $CaCO_3$ was necessary prior to analysis of BOM by IRMS. We selected a total of 67 BOM samples from 42 lakes and removed inorganic carbon by acid fumigation [36]. Results showed only minimal changes in $\delta^{13}C_{BOM}$ after acid fumigation. The mean difference of $\delta^{13}C_{BOM}$ untreated $- \delta^{13}C_{BOM}$ fumigated was $= -0.05$‰, range ($-0.23$ to $+0.07$‰), n = 67); due to this small difference in lakes, we used untreated $\delta^{13}C_{BOM}$ samples. In estuaries, $\delta^{13}C_{BOM}$ from three stations showed large decreases in $\delta^{13}C_{BOM}$ after acid fumigation. Therefore, in estuaries we used results from acid fumigated $\delta^{13}C_{BOM}$ samples for comparisons.

In lakes, the means of $\delta^{15}N_{UN}$ and $\delta^{13}C_{UN}$ or individual values (where only one organism was found) were used for the comparisons with $\delta^{15}N_{BOM}$ and $\delta^{13}C_{BOM}$ and with land cover variables and population. In estuaries the mean values of the triplicate $\delta^{15}N_{BOM}$ and $\delta^{13}C_{BOM}$ ratios were compared with the corresponding mean $\delta^{15}N_{MM}$ and $\delta^{13}C_{MM}$ ratios, or with individual values where only one bivalve was found, at a total of 28 stations from nine estuaries. Only one station was sampled in Narragansett Bay, but the nitrogen and carbon isotope ratios in this station, $\delta^{15}N_{BOM} = 7.8$‰ $\pm$ 0.5‰ SD; $\delta^{13}C_{BOM} = -18.3$‰ $\pm$ 0.3‰ SD and $\delta^{15}N_{MM} = 13.0$‰ $\pm$ 0.2‰ SD; $\delta^{13}C_{MM} = -17.4$‰ $\pm$ 0.2‰ SD, agreed with summary values for Narragansett Bay calculated from other sources. These sources measured nitrogen and carbon isotopes in the surface sediment (top 2 cm) of core samples from the axis of the Bay and found $\delta^{15}$N = 7.9‰ $\pm$ 0.35‰ SD, and $\delta^{13}C_{BOM} = -19.9$‰ $\pm$ 0.84‰ SD [37]. The *M. mercenaria* samples were from stations taken throughout the Bay and had means of $\delta^{15}N_{MM} = 13.2$‰ $\pm$ 0.54‰ SD, and $\delta^{13}C_{MM} -16.76 \pm 0.61$‰ SD [38]. The grand means of $\delta^{15}N_{BOM}$ and $\delta^{15}N_{MM}$ from all stations within each of the nine estuaries were used for comparison with land cover and population variables (*n* = 9).

For comparisons of $\delta^{15}$N ratios in filter feeding bivalves with those in BOM or surface sediment, we developed two relationships: (1) $\delta^{15}$N in mussels ($\delta^{15}N_{UN}$) as a function of $\delta^{15}N_{BOM}$ in the 26 lakes, and (2) $\delta^{15}$N in hard-shell clams ($\delta^{15}N_{MM}$) as a function of $\delta^{15}N_{BOM}$ from a total of 28 stations from nine estuaries. These regressions were compared using analysis of covariance (ANCOVA). Similar relationships with $\delta^{13}C_{UN}$ as a function of $\delta^{13}C_{BOM}$, and $\delta^{13}C_{MM}$ as a function of $\delta^{13}C_{BOM}$ were developed and analyzed.

Land cover data within the watersheds of lakes and estuaries were obtained from RIGIS (2003–2004) [39] and Mass GIS (2005) [40] imagery with 0.6 m and 0.5 m resolution, respectively. These data were categorized using the land use designations of urban/residential, agricultural, forest, wetland, barren, and water, and percentages of each land group were calculated. Percent impervious surface in watersheds was calculated separately within the watersheds of lakes and estuaries obtained from RIGIS (2007) [39] and Mass GIS (2005) [40]. Population estimates for 2010 were made using dasymetric rasters developed using a USGS tool [41] (Sleeter and Gould 2007, updated 2016) with data from United States Census Bureau [42] (2010). Land use data were from RIGIS (2011) [39] for Rhode Island and from Mass GIS (2005) [40] for Massachusetts.

We compared Spearman and Pearson correlations of land cover variables and population in watersheds with $\delta^{15}N_{BOM}$, $\delta^{13}C_{BOM}$ in 51 lakes and in the nine estuaries. Similar comparisons were also made using only data for $\delta^{15}N_{BOM}$, $\delta^{13}C_{BOM}$ and with $\delta^{15}N_{UN}$, $\delta^{13}C_{UN}$ in the 26 lakes where mussels were found. Comparisons were also made between $\delta^{15}N_{BOM}$ and $\delta^{15}N_{MM}$ and between $\delta^{13}C_{BOM}$ and $\delta^{13}C_{MM}$ in the nine estuaries. Correlations used individual $\delta^{15}N_{BOM}$ and $\delta^{13}C_{BOM}$ values and site mean values of $\delta^{15}N_{UN}$ and $\delta^{13}C_{UN}$ for lakes and grand means of $\delta^{15}N_{BOM}$, $\delta^{13}C_{BOM}$, $\delta^{15}N_{MM}$, and $\delta^{13}C_{MM}$ from all stations within each estuary. We compared the results from both correlation methods

to determine the extent to which the presence of non-normality of some variables in the Pearson correlations impacted results.

We compared relationships of land cover variables and population with each other and selected log % impervious surface as the most appropriate variable for representing anthropogenic land development. We developed regression relationships with log % impervious surface as a function of $\delta^{15}N_{BOM}$ from the lakes and estuaries.

Correlations, least squares regressions, ANCOVA, and regression model selection were done using SAS software version 9.4 (SAS Institute Cary, NC, USA). Student's *t*-tests, F-tests for equality of variance, and regression analyses were done using Microsoft Excel. We reported results of Pearson correlation coefficients with significance indicated for pairwise comparisons to show relationships within the data. The level of significance for statistical tests was $p \leq 0.05$.

## 3. Results

### 3.1. Comparisons of Bivalves with BOM

Increased $\delta^{15}N$ in mussels ($\delta^{15}N_{UN}$) from lakes and increased $\delta^{15}N$ in hard shell clams ($\delta^{15}N_{MM}$) from estuaries were associated with greater $\delta^{15}N_{BOM}$ ($R^2 = 0.82$ and $p < 0.001$ for both; Figure 1). The slopes of these relationships were similar in lakes and estuaries (ANCOVA, $p = 0.12$), but their intercepts were significantly different ($p < 0.001$). Increased $\delta^{13}C_{UN}$ were associated with greater $\delta^{13}C_{BOM}$ in lakes ($R^2 = 0.70$, $p < 0.001$), but $\delta^{13}C_{MM}$ was not significantly correlated with $\delta^{13}C_{BOM}$ in estuaries (Figure 2).

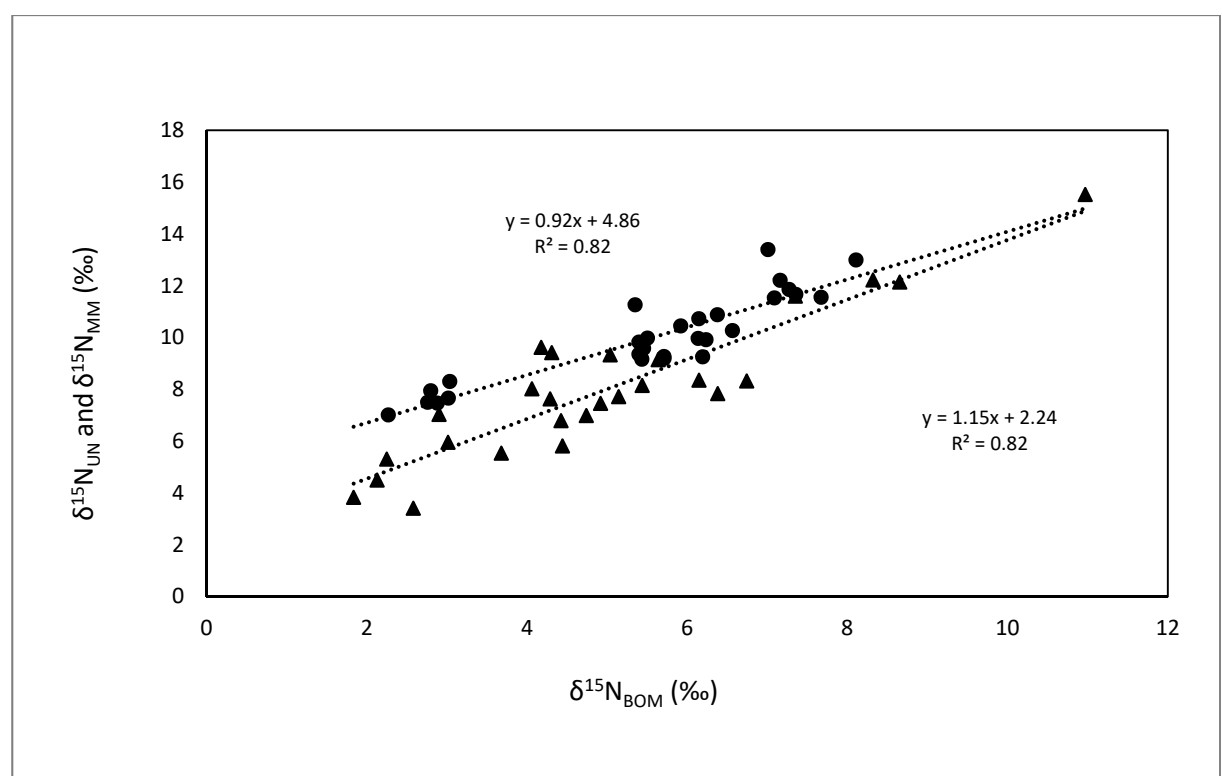

**Figure 1.** Regression comparisons of $\delta^{15}N$ in unionid (UN) mussels ($\delta^{15}N_{UN}$) from 26 lakes (▲), and in hard-shell clams, *M. mercenaria* (MM) ($\delta^{15}N_{MM}$) from a total of 28 stations from nine estuaries (●), as functions of $\delta^{15}N$ in sediment benthic organic material (BOM), ($\delta^{15}N_{BOM}$). $\delta^{15}N_{UN} = 1.15 \delta^{15}N_{BOM} + 2.24$, $R^2 = 0.82$, n = 26, $p < 0.001$. (2) $\delta^{15}N_{MM} = 0.92 \delta^{15}N + 4.86$, $R^2 = 0.82$, n = 28, $p < 0.001$.

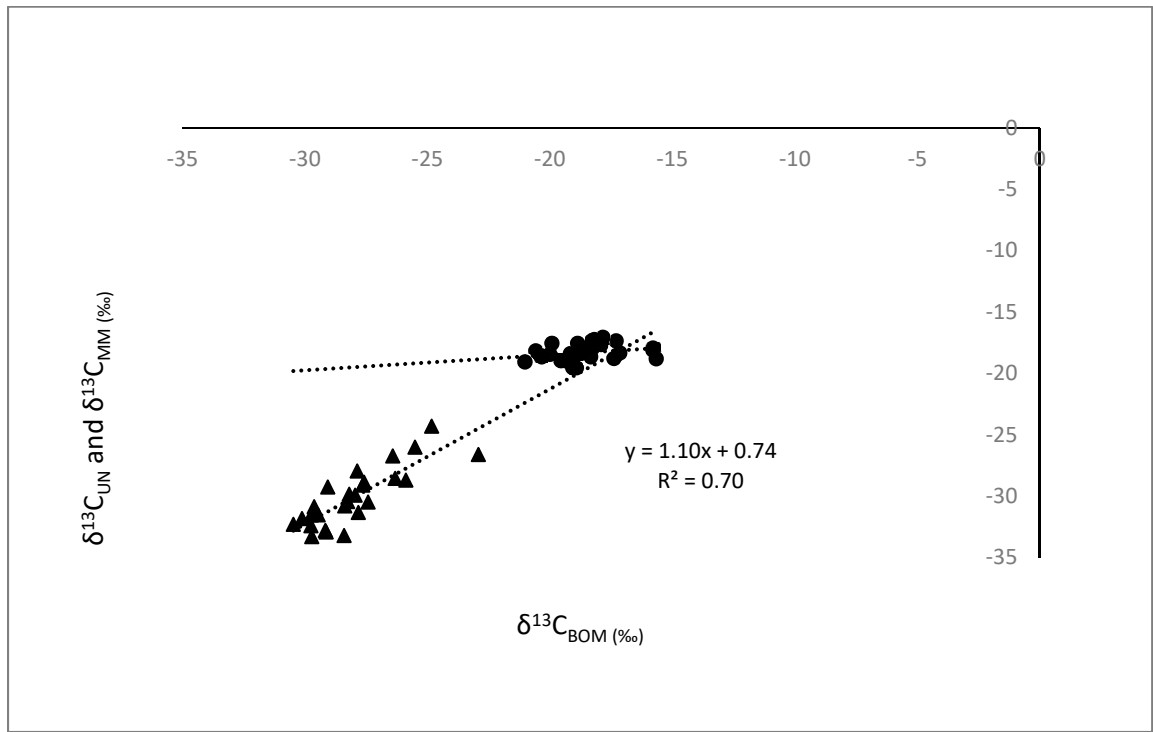

**Figure 2.** Regression comparisons of $\delta^{13}$C in unionid (UN) mussels ($\delta^{13}$C$_{UN}$) from 26 lakes (▲), and in hard-shell clams, *M. mercenaria* (MM) ($\delta^{13}$C$_{MM}$) from a total of 28 stations from nine estuaries (●), as functions of $\delta^{13}$C in sediment benthic organic material (BOM), ($\delta^{13}$C$_{BOM}$). $\delta^{13}$C$_{UN}$ = 1.10 $\delta^{13}$C$_{BOM}$ + 0.74, $R^2$ = 0.70, n = 26, *p* < 0.001. The $\delta^{13}$C$_{MM}$ regression with $\delta^{13}$C$_{BOM}$ in estuaries was not significant.

### 3.2. Comparisons of BOM with Land Cover and Population

We found a general agreement of the Pearson and Spearman correlations of $\delta^{15}$N$_{BOM}$ with land cover variables and population in watersheds surrounding lakes and estuaries which indicated that the lack of normality in some variables in the Pearson correlations did not adversely impact results (Table 1). In the watersheds of 51 lakes, we found $\delta^{15}$N$_{BOM}$ was significantly positively correlated with % urban residential land, log % impervious surface and log population and significantly negatively correlated with % forest. Lower correlation coefficients of $\delta^{15}$N$_{BOM}$ were found with % wetland, % water, and % agriculture and these correlations were not significant (Table 1). The correlation of $\delta^{15}$N$_{BOM}$ with % barren land, which represented both % barren land and % beaches, showed a significant, but low correlation. A review of the % barren land variable showed 28 of the 51 lakes had % barren land = 0; therefore, the significant correlation was likely an artifact.

In the watersheds of nine estuaries, correlations were not as strong as in the lakes, and significant Pearson correlations were found only for $\delta^{15}$N$_{BOM}$ with log % impervious surface. The variable % forest in watersheds of estuaries was not significant as it was in the lakes, but this may have resulted because of the large extent of salt marsh grasses surrounding several of the estuarine sites and the consequent decreases in % forest. The variables % wetland and % water showed significant correlations with $\delta^{15}$N$_{BOM}$ in estuaries, but this resulted because of the large extent of wetland and ponds surrounding one undeveloped site, Sage Lot Pond, which had the lowest $\delta^{15}$N$_{BOM}$ value and the highest percentages of wetland and water. When the percentages of wetland and water for Sage Lot Pond were removed from the correlations, as in Table 1, their correlations with $\delta^{15}$N$_{BOM}$ became non-significant.

**Table 1.** Comparisons of Spearman and Pearson correlations of $\delta^{15}$N in sediment benthic organic material (BOM), $\delta^{15}$N$_{BOM}$ with land cover and population variables in watersheds of 51 lakes and nine estuaries. Land cover variables included log % impervious surface (Log%Impsur) which was calculated separately, % urban residential (% UrRes), log population (Logpop), % Forest, % Agriculture (% Agr), % Wetland (% WetL), % Barren, and % Water.

| | Lakes | | | | | | | |
|---|---|---|---|---|---|---|---|---|
| | **Spearman Correlations** | | | | | | | |
| | Log%Impsur | % UrRes | Logpop | % Forest | % Agr | % WetL | % Barren | % Water |
| $\delta^{15}$N$_{BOM}$ | 0.81 | 0.78 | 0.54 | −0.71 | 0.15 | −0.26 | 0.31 | −0.18 |
| *p* | <0.001 | <0.001 | <0.001 | <0.001 | NS | NS | 0.03 | NS |
| | **Pearson Correlations** | | | | | | | |
| | Log%Impsur | % UrRes | Logpop | % Forest | % Agr | % WetL | % Barren | % Water |
| $\delta^{15}$N$_{BOM}$ | 0.79 | 0.69 | 0.59 | −0.71 | 0.14 | −0.23 | 0.35 | −0.27 |
| *p* | <0.001 | <0.001 | <0.001 | <0.001 | NS | NS | 0.01 | NS |
| | Estuaries | | | | | | | |
| | **Spearman Correlations** | | | | | | | |
| | Log%Impsur | %UrRes | Logpop | % Forest | % Agr | % WetL | % Barren | % Water |
| $\delta^{15}$N$_{BOM}$ | 0.73 | 0.5 | 0.33 | 0.02 | 0.25 | −0.17 | −0.57 | 0.05 |
| *p* | 0.02 | NS | NS | NS | NS | NS | NS | NS |
| | **Pearson Correlations** | | | | | | | |
| | Log%Impsur | % UrRes | Logpop | % Forest | % Agr | % WetL | % Barren | % Water |
| $\delta^{15}$N$_{BOM}$ | 0.75 | 0.48 | 0.67 | 0.04 | 0.31 | −0.03 | −0.13 | −0.07 |
| *p* | 0.02 | NS | 0.05 | NS | NS | NS | NS | NS |

NS indicates pair- wise correlation was not significant.

In the 26 lakes with mussels, significant positive Pearson correlations were found for both $\delta^{15}$N$_{BOM}$ and $\delta^{15}$N$_{UN}$ with log % impervious surface, % urban residential and log population, and significant negative correlations were found with % forest (Table 2). Similar correlations of $\delta^{15}$N$_{BOM}$ and $\delta^{15}$N$_{MM}$ with land cover and population in watersheds of the estuaries showed a significant correlation only for $\delta^{15}$N$_{BOM}$ with log % impervious surface (Table 2).

**Table 2.** Pairwise Pearson correlation coefficients for $\delta^{15}$N in sediment benthic organic material ($\delta^{15}$N $_{BOM}$) and $\delta^{15}$N in the bivalves unionid mussels ($\delta^{15}$N$_{UN}$) and *M. mercenaria* ($\delta^{15}$N$_{MM}$) with log percent impervious surface (Log%Impsur), which was calculated separately, % urban residential land (% UrRes), log population (Logpop) and % Forest in watersheds of 26 lakes and nine estuaries.

| | Lakes | | | |
|---|---|---|---|---|
| | Log%Impsur | % UrRes | Logpop | % Forest |
| $\delta^{15}$N$_{BOM}$ | 0.79 | 0.74 | 0.49 | −0.73 |
| *p* | <0.001 | <0.001 | 0.011 | <0.001 |
| $\delta^{15}$N$_{UN}$ | 0.75 | 0.66 | 0.63 | −0.67 |
| *p* | <0.001 | <0.001 | <0.001 | <0.001 |
| | Estuaries | | | |
| | Log%Impsur | % UrRes | Logpop | % Forest |
| $\delta^{15}$N$_{BOM}$ | 0.75 | 0.48 | 0.67 | 0.04 |
| *p* | 0.019 | NS | NS | NS |
| $\delta^{15}$N$_{MM}$ | 0.61 | 0.36 | 0.55 | 0.19 |
| *p* | NS | NS | NS | NS |

NS indicates pair- wise correlation was not significant.

Our comparisons of regressions of land cover and population variables with each other and with $\delta^{15}$N$_{BOM}$ showed the highest coefficients of determination with log % impervious surface, which we used for representing anthropogenic land development and compared it with $\delta^{15}$N$_{BOM}$ in the lakes and estuaries. In lakes we found a significant regression of log %

impervious surface as a function of $\delta^{15}N_{BOM}$ in the watersheds (Figure 2). An empirical equation was derived from this regression (Equation (1)).

$$\log \% \text{ impervious surface} = 0.16 \, (\delta^{15}N_{BOM}) + 0.16, R^2 = 0.62, p < 0.001, n = 51 \quad (1)$$

Within the estuaries we found a significant regression of log % impervious surface as a function of $\delta^{15}N_{BOM}$ in the watersheds (Figure 3 and Equation (2)). To determine how the removal of the much larger Narragansett Bay from the regression would impact the relationship we omitted it from a second regression analysis. The regression without the Narragansett Bay watershed was only slightly different (Equation (3)).

$$\log \% \text{ impervious surface} = 0.05 \, (\delta^{15}N_{BOM}) + 0.82, R^2 = 0.57, p = 0.019, n = 9 \quad (2)$$

$$\log \% \text{ impervious surface} = 0.056 \, (\delta^{15}N_{BOM}) + 0.79, R^2 = 0.60, p = 0.02, n = 8 \quad (3)$$

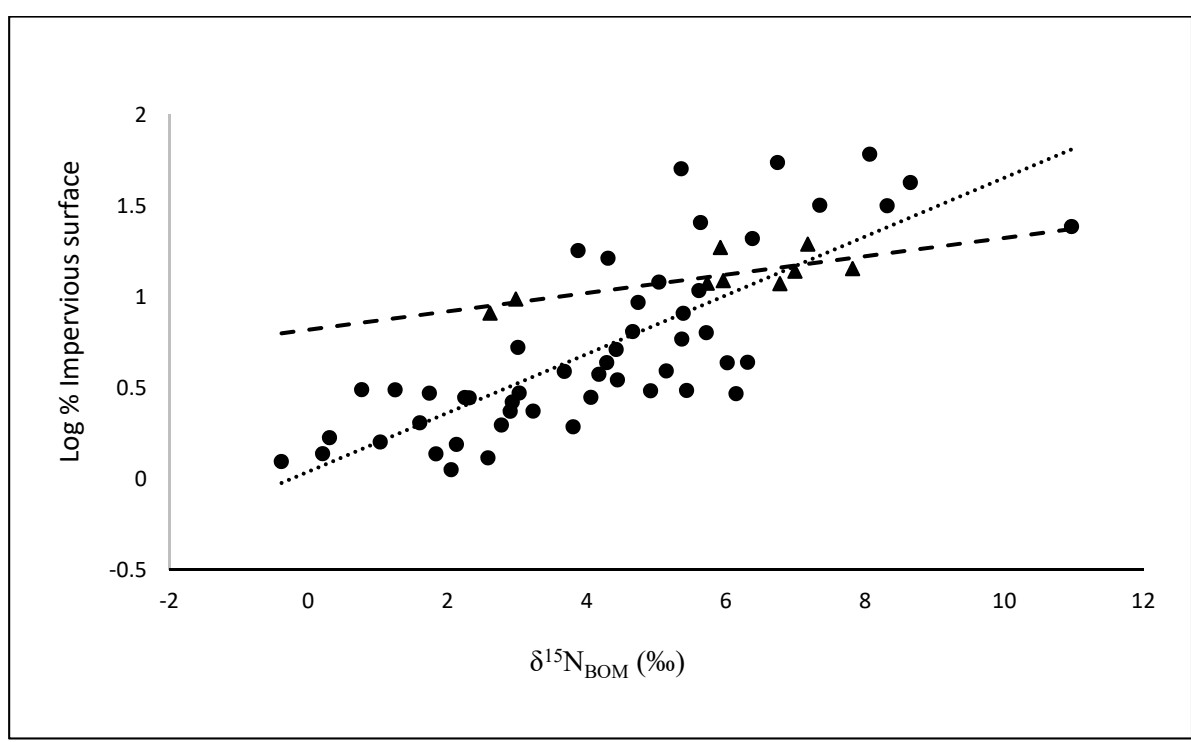

**Figure 3.** Regression comparisons of log % impervious surface in watersheds of 51 lakes (dotted line) and nine estuaries (dashed line) as functions of $\delta^{15}N$ in sediment benthic organic material (BOM), $\delta^{15}N_{BOM}$. Equations for lines are log % impervious surface = 0.16 $(\delta^{15}N_{BOM})$ + 0.04, $R^2$ = 0.62, $p < 0.001$, in 51 lakes (●), and log % impervious surface = 0.05 $(\delta^{15}N_{BOM})$ = 0.82, $R^2$ = 0.57, $p = 0.02$, in nine estuaries (▲).

Comparisons of $\delta^{13}C_{BOM}$ with land cover and population variables showed no significant correlations for watersheds of lakes, or estuaries.

## 4. Discussion

For environmental assessment of how anthropogenic activities in watersheds affect lakes and estuaries, our results suggest that a sediment-based method presents a useful alternative to mussels and other organisms, which commonly can be absent from sites. In the present study, we found a statistical similarity of slopes and $R^2$ values (0.82) of regression relationships between $\delta^{15}N_{UN}$ as a function of $\delta^{15}N_{BOM}$ in lakes and $\delta^{15}N_{MM}$ as a function of $\delta^{15}N_{BOM}$ in estuaries. This level of agreement indicates that $\delta^{15}N_{BOM}$ from sediment closely duplicates $\delta^{15}N$ found in bivalves, which are often considered

primary consumers and have been used in environmental assessments [22–24,43]. We found different intercepts of regressions between the lakes and the estuaries. The different intercepts may indicate variation in accumulation through feeding or metabolism by the different species of bivalves present in lakes versus those of estuaries. Studies indicate the dietary components of mussels are poorly understood [44,45], and species differences in $\delta^{15}N$ of mussels taken from the same sites have been reported [24,46]. Additionally, species-specific differences of $\delta^{15}N$ in hard shell clams and in soft shell clams (*Mya Arenaria*) collected from the same small estuaries on Cape Cod, USA have been found [26].

In lakes of the present study, the relationship of $\delta^{13}C_{UN}$ as a function of $\delta^{13}C_{BOM}$ was significant; however, the relationship of $\delta^{13}C_{MM}$ from estuaries as a function of $\delta^{13}C_{BOM}$ was not significant. The diet of M. Mercenaria is planktonic microalgae [47], whereas the diets of mussels are more varied as cited above [45,46]. These dietary differences may be the reason for the relatively small range of $\delta^{13}C_{MM}$ relative to $\delta^{13}C_{UN}$. Our findings indicate that although $\delta^{15}N_{BOM}$ could be used as a surrogate for bivalves in assessments of nitrogen concentrations and impacts in lakes and estuaries, $\delta^{13}C_{BOM}$ appears to have utility as a surrogate for $\delta^{13}C$ only for bivalves in lakes.

Our comparisons of correlations of $\delta^{15}N_{BOM}$ with variables indicative of anthropogenic land development of watersheds surrounding lakes showed significant positive correlations with log population, log % impervious surface, % urban residential, and a negative correlation with % forest. These results support the use of $\delta^{15}N_{BOM}$ for assessments in watersheds surrounding lakes. Due to the dynamic nature of estuaries, it is often difficult to assess their environmental condition.

Environmental assessments in estuaries have used a variety of methods to develop and refine benthic organism-based indices to evaluate ecological status of sites [48–50]. However, collection and identification of species present in estuarine sediment samples are labor intensive and subject to variability depending upon seasonality and site-specific variables (e.g., sediment type). Results of analyses of nutrients and other aquatic variables also have been used to assess and rank aquatic site condition, as in National coastal condition assessments [3]. These assessments have been highly useful for summarizing data, but they do not link nutrients directly to ecological responses and nutrient concentrations are inherently variable over time. Thus, the addition of a time integrative measure such as $\delta^{15}N_{BOM}$ may be useful for these assessments.

The correlations of $\delta^{15}N_{BOM}$ with land use and population variables in estuarine watersheds were not as strong as in lakes, and $\delta^{15}N_{BOM}$ was only significantly correlated with log % impervious surface. These results likely reflect the highly dynamic nature of the estuarine environment and the influence of freshwater and seawater inputs, tides, complex flows, and variable marine algal (food) sources on the composition of the organic material in sediments and therefore on $\delta^{15}N_{BOM}$ in estuarine sediments, but still show that $\delta^{15}N_{BOM}$ in estuaries is associated with watershed development. One of our estuaries, Narragansett Bay, has a watershed which is ~38 times larger than the next largest estuarine watershed, and has been found to be highly influenced by nutrients from wastewater [38]. To determine whether the inclusion of Narragansett Bay watershed had an over whelming influence on the regression, we removed it from the data set and recalculated the regression. We found almost no change in the regression equation which indicates that the regression is relatively robust and provides further support for using $\delta^{15}N_{BOM}$ to indicate effects of anthropogenic land development in watersheds of estuaries.

In other studies, the sources of nitrogen inputs to aquatic sites have been found to consist of human or livestock wastes, inputs of human wastewater, or from application of nitrogen in fertilizers in surrounding watersheds. In the present study, the percentage of agricultural land use in watersheds surrounding lakes and estuaries was low and did not represent a significant variable for predicting $\delta^{15}N$ in bivalves or $\delta^{15}N_{BOM}$. Our results showed that variables indicating anthropogenic land development (i.e., increased human population, urban/residential development, percentage of impervious surface and decreased forest) were principal explanatory variables and likely indicated the consequent

input of wastewater to sites. One study suggested that drivers of $\delta^{15}N$ through nitrogen inputs in lakes may show regional differences [51]. Our study sites were all in New England, and our results coincided with anthropogenic development in watersheds. We could not test regional differences but acknowledge that regional differences in nitrogen sources and inputs to aquatic sites are likely [25,52].

The lakes and estuaries of the present study, except for Narragansett Bay, are relatively small; therefore, land use in the surrounding watersheds was more likely to have been reflected in the $\delta^{15}N_{BOM}$. However, in larger waterbodies comparisons of $\delta^{15}N_{BOM}$ with anthropogenic activities in watersheds may be influenced by a large range of activities and inputs which are difficult to quantify. Therefore, in large lakes and estuaries, studies to determine the appropriate sampling strategies for using stable isotopes in sediment will be needed for development of adequate assessments.

Overall, the present study showed highly significant positive relationships between $\delta^{15}N$ in bivalves and BOM, which indicated that $\delta^{15}N_{BOM}$ may be used as a surrogate for $\delta^{15}N$ in bivalve mollusks. Our findings suggest that $\delta^{15}N_{BOM}$ provides an effective summary metric for assessing and ranking the environmental conditions of lakes and estuaries resulting from anthropogenic land development of watersheds.

**Supplementary Materials:** The following supporting information can be downloaded at: https://www.mdpi.com/article/10.3390/w14223712/s1, Table S1: Location data for study lakes and watershed size (ha) for lakes and estuaries; Table S2: Number, size data, δ15N and δC13 for Mussels (UN) from 26 Lakes; Table S3: Number, size data, δ15N and δC13 and sample site locations for Mercenaria mercenaria from a total of Estuaries 28 stations from nine Estuaries.

**Author Contributions:** Conceptualization, J.L.L., J.R.S., A.K. and N.J.S.; Data curation, J.R.S.; Formal analysis, J.L.L.; Investigation, J.R.S.; Methodology, J.L.L. and J.R.S.; Project administration, J.L.L.; Supervision, J.L.L. and A.K.; Visualization, J.R.S., A.K. and M.C.; Writing—original draft, J.L.L.; Writing—review and editing, J.R.S., A.K., M.C. and N.J.S. All authors have read and agreed to the published version of the manuscript.

**Funding:** Michael Charpentier was funded through a contract with General Dynamics Information Technology. The other authors were funded by the U.S. Environmental Protection Agency.

**Institutional Review Board Statement:** This manuscript has been through ACESD review procedures and has been cleared for publication.

**Informed Consent Statement:** Not applicable.

**Data Availability Statement:** Data is located in: https://sciencehub.epa.gov/sciencehub/datasets/3803 (accessed 10 November 2022).

**Acknowledgments:** The authors thank Cathleen Wigand, Michaela Cashman, Laura Coiro, Betty Kreakie and Marty Chintala for their expertise and helpful technical reviews of this manuscript.

**Conflicts of Interest:** The authors declare no conflict of interest. This manuscript has been subjected to Agency review and has been approved for publication. The views expressed in this article are those of the authors and do not necessarily represent the views or policies of the U.S. Environmental Protection Agency. Any mention of trade names, products, or services does not imply an endorsement by the U.S. Government or the U.S. Environmental Protection Agency. The EPA does not endorse any commercial products, services, or enterprises.

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
