# Peer review of "Evaluation of Nitrogen and Carbon Stable Isotopes in Filter Feeding Bivalves and Surficial Sediment for Assessing Aquatic Condition in Lakes and Estuaries"

_water, doi:10.3390/w14223712_

Round 1

Reviewer 1 Report

The article under review entitled "Evaluation of nitrogen and carbon stable isotopes in filter feeding bivalves and surficial sediment for assessing aquatic condition in lakes and estuaries" presents the results of research on the use of the isotope composition of benthic organic matter as an "easier" indicator of anthropogenic impact on water ecosystems (lakes and estuaries). This indicator would be an alternative for isotope ratios of filter feeding bivalves. In my opinion, the article is correctly written. The research and analysis methods are described in detail, the research results clearly presented and properly discussed. However, I suppose that the conclusions formulated by the authors may not be confirmed for other types of catchments, especially those that are heavily influenced by agriculture. Nevertheless, I think the paper should be published in the "Water" journal with a minor improvements (in comments) as the results presented will provide a good basis for further research.

Comments:

The article does not contain a map with the location of the studied lakes and estuaries, therefore I suggest that in Supplemental Table 1 provide the geographical coordinates of these objects or sampling sites

Figures should be numbered consecutively 1, 2, 3 and not 1a, 1b, 2

Figs. 1a, 1b and 2 - please add axis markers and units of  d15N and d13C

Please complete the relevant information on lines from 388 to 394

Supplemental Table 1 - "ha" instead of "HA"

Author Response

Thank you for your careful and thoughtful review.

We have included locations of sampled lakes in Supplemental Table 1. Sample location in estuaries are in Supplemental Table 3. 

We have numbered Figures as requested.

We have added equations and units to the axes to graphical figures.

 We have included the relevant information in 388 to 394.

We have changes HA to ha in Supplemental Table 1.

Reviewer 2 Report

This is an interesting study and well-written paper. Please add Author Contributions and Data Availability Statement after the main text.

Author Response

Thank you for your review.

We have added the the authors contributions and data availability statements as requested.

Reviewer 3 Report

This study tested the use of stable isotopes of nitrogen 15N) and carbon (δ13C) in benthic organic matter (BOM) as proxies for isotope ratios of filter feeding bivalves in lakes and estuaries, and evaluated if stable isotope ratios in benthic organic matter could be used to assess impacts from anthropogenic land development of watersheds. The result of research showed highly significant positive relationships between δ15N in bivalves and BOM, which indicated that δ15NBOM may be used as a surrogate for δ15N in bivalve mollusks. The findings suggest that δ15NBOM provides an effective summary metric for assessing and ranking the environmental conditions of lakes and estuaries resulting from anthropogenic land development of watersheds.

The experimental methods and statistical methods of this study are very rigorous. For example, pre-experiments were conducted before acidification to compare sediments from different regions to see if acidification was required, and differences in Pearson and Spearman correlations were compared. The manuscript has been already well written and I have some minor comments and suggestions as follows:

1.       There are many aquatic ecosystems, such as rivers, lakes, estuaries and coastal zones. Why did this paper choose lakes and estuaries for research?

2.       Line115-127:It would be better to add a sampling site diagram here.

3.       Page6: Legend and regression equation can be added to the figure to make it more intuitive for readers.

4.       Line325-329: BOM are generally considered to be one of the main food sources of benthic shellfish. In this study, the relationship of δ13CUN as a function of δ13CBOM was significant; however, the relationship of δ13CMM from estuaries as a function of δ13CBOM was not significant. Why did this happen? The author did not describe this. I think it would be better to talk more about that, for example, is it related to the fact that the basic food sources in estuaries are more complex?

5.       Line376-382: I agree with the author here. Isotopes may differ markedly over large spatial ranges. Therefore, in large lakes and estuaries, studies to determine the appropriate sampling strategies for using stable isotopes in sediment will be needed for development of adequate assessments.

Author Response

Comment #1 We have studies underway to test the utility of stable isotopes in sediment for assessing stream condition. The present work provides a useful beginning to future work in other aquatic systems.

Comment#2 We have added coordinates for the lakes and for the sampling sites in the estuaries in the supplemental Tables.

Comment #3 We have added equations to the Figures. The legends also contain the equations for the figures.

Comment #4 The range of d13C in mercenaria mercenaria is relatively small. This may be because their diet is planktonic microalgae, whereas the diets of mussels are more varied and rather poorly understood. We have included a reference to this in the revision.

Comment # 5 In future work we will assess the use of our techniques in larger studies e. g. USEPA National Coastal Condition Assessment (NCCA).
